# Automatically Learning Compact Quality-aware Surrogates for Optimization Problems

**Kai Wang**
Harvard University
Cambridge, MA
kaiwang@g.harvard.edu

**Bryan Wilder**
Harvard University
Cambridge, MA
bwilder@g.harvard.edu

**Andrew Perrault**
Harvard University
Cambridge, MA
aperrault@g.harvard.edu

**Milind Tambe**
Harvard University
Cambridge, MA
milind_tambe@harvard.edu

## Abstract

Solving optimization problems with unknown parameters often requires learning a predictive model to predict the values of the unknown parameters and then solving the problem using these values. Recent work has shown that including the optimization problem as a layer in the model training pipeline results in predictions of the unobserved parameters that lead to higher decision quality. Unfortunately, this process comes at a large computational cost because the optimization problem must be solved and differentiated through in each training iteration; furthermore, it may also sometimes fail to improve solution quality due to non-smoothness issues that arise when training through a complex optimization layer. To address these shortcomings, we learn a low-dimensional surrogate model of a large optimization problem by representing the feasible space in terms of meta-variables, each of which is a linear combination of the original variables. By training a low-dimensional surrogate model end-to-end, and jointly with the predictive model, we achieve: i) a large reduction in training and inference time; and ii) improved performance by focusing attention on the more important variables in the optimization and learning in a smoother space. Empirically, we demonstrate these improvements on a non-convex adversary modeling task, a submodular recommendation task and a convex portfolio optimization task.

## 1 Introduction

Uncertainty is a common feature of many real-world decision-making problems because critical data may not be available when a decision must be made. Here is a set of representative examples: recommender systems with missing user-item ratings [21], portfolio optimization where future performance is uncertain [29], and strategic decision-making in the face of an adversary with uncertain objectives [24]. Often, the decision-maker has access to features that provide information about the values of interest. In these settings, a *predict-then-optimize* [12] approach naturally arises, where we learn a model that maps from the features to a value for each parameter and optimize using this point estimate [44]. In principle, any predictive modeling approach and any optimization approach can be applied, but using a generic loss function to train the model may result in poor decision performance. For example, a typical ratings prediction approach in recommendation system may equally weight errors across different items, but in the recommendation task, misclassifying a trendy item can result in more revenue loss than misclassifying an ordinary item. We may instead

want to train our model using a "task-based" or "decision-focused" loss, approximating the decision quality induced by the predictive model, which can be done by embedding the optimization problem as a layer in the training pipeline. This end-to-end approach improves performance on a variety of tasks [5, 43, 8].

Unfortunately, this end-to-end approach suffers from poor scalability because the optimization problem must be solved and differentiated through on every training iteration. Furthermore, the output of the optimization layer may not be smooth, sometimes leading to instabilities in training and consequently poor solution quality. We address these shortcomings that arise in the end-to-end approach due to the presence of a complex optimization layer by replacing it with a simpler surrogate problem. The surrogate problem is learned from the data by automatically finding a reparameterization of the feasible space in terms of meta-variables, each of which is a linear combination of the original decision variables. The new surrogate problem is generally cheaper to solve due to the smaller number of meta-variables, but it can be lossy—the optimal solution to the surrogate problem may not match the optimal solution to the original. Since we can differentiate through the surrogate layer, we can optimize the choice of surrogate together with predictive model training to minimize this loss. The dimensionality reduction offered by a compact surrogate simultaneously reduces training times, helps avoid overfitting, and sometimes smooths away bad local minima in the training landscape.

In short, we make several contributions. First, we propose a linear reparameterization scheme for general optimization layers. Second, we provide theoretical analysis of this framework along several dimensions: (i) we show that desirable properties of the optimization problem (convexity, submodularity) are retained under reparameterization; (ii) we precisely characterize the tractability of the end-to-end loss function induced by the reparameterized layer, showing that it satisfies a form of coordinate-wise quasiconvexity; and (iii) we provide sample complexity bounds for learning a model which minimizes this loss. Finally, we demonstrate empirically on a set of three diverse domains that our approach offers significant advantages in both training time and decision quality compared previous approaches to embedding optimization in learning.

**Related work**   Surrogate models [15, 36, 27] are a classic technique in optimization, particularly for black-box problems. Previous work has explored linear reparameterizations to map between low and high fidelity models of a physical system [4, 37, 3] (e.g., for aerospace design problems). However, both the motivation and underlying techniques differ crucially from our work: previous work has focused on designing surrogates by hand in a domain-specific sense, while we leverage differentiation through the optimization problem to automatically produce a surrogate that maximizes overall decision quality.

Our work is closest to the recent literature on differentiable optimization. Amos et al. [2] and Agrawal et al. [1] introduced differentiable quadratic programming and convex programming layers, respectively, by differentiating through the KKT conditions of the optimization problem. Donti et al. [8] and Wilder et al. [43] apply this technique to achieve end-to-end learning in convex and discrete combinatorial programming, respectively. Perrault et al. [32] applied the technique to game theory with a non-convex problem, where a sampling approach was proposed by Wang et al. [41] to improve the scalability of the backward pass. All the above methods share scalability and non-smoothness issues: each training iteration requires solving the entire optimization problem and differentiating through the resulting KKT conditions, which requires $O(n^3)$ time in the number of decision variables and may create a non-smooth objective. Our surrogate approach aims to rectify both of these issues.

## 2   Problem Statement

We consider an optimization problem of the form: $\min_{\mathbf{x} \text{ feasible}} f(\mathbf{x}, \theta_{\text{true}})$. The objective function depends on a parameter $\theta_{\text{true}} \in \Theta$. If $\theta_{\text{true}}$ were known, we assume that we could solve the optimization problem using standard methods. We consider the case that parameter $\theta_{\text{true}}$ is unknown and must be inferred from the given available features $\xi$. We assume that $\xi$ and $\theta_{\text{true}}$ are correlated and drawn from a joint distribution $\mathcal{D}$, and our data consists of samples from $\mathcal{D}$. Our task is to select the optimal decision $\mathbf{x}^*(\xi)$, function of the available feature, to optimize the expected objective value:

$$\min_{\mathbf{x}^* \text{ feasible}} E_{(\xi, \theta_{\text{true}}) \sim \mathcal{D}}[f(\mathbf{x}^*(\xi), \theta_{\text{true}})] \tag{1}$$

In this paper, we focus on a *predict-then-optimize* [12, 10] framework, which proceeds by learning a model $\Phi(\cdot, w)$, mapping from the features $\xi$ to the missing parameter $\theta_{\text{true}}$. When $\xi$ is given, we first

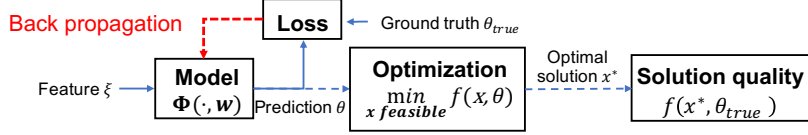

Figure 1: Two-stage learning back-propagates from the loss to the model, ignoring the latter effect of the optimization layer.

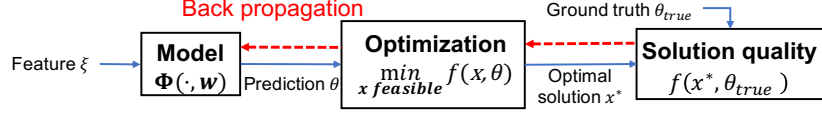

Figure 2: End-to-end decision-focused learning back-propagates from the solution quality through the optimization layer to the model we aim to learn.

infer $\theta = \Phi(\xi, w)$ and then solve the resulting optimization problem to get the optimal solution $\mathbf{x}^*$:

$$\min_{\mathbf{x}} \quad f(\mathbf{x}, \theta), \quad \text{s.t.} \quad h(\mathbf{x}) \leq 0, \quad A\mathbf{x} = b \tag{2}$$

This reduces the decision-making problem with unknown parameters to a predictive modeling problem: how to learn a model $\Phi(\cdot, w)$ that leads to the best performance.

A standard approach to solve the predict-then-optimize problem is *two-stage* learning, which trains the predictive model without knowledge of the decision-making task (Figure 1). The predictive model minimizes the mismatch between the predicted parameters and the ground truth: $E_{(\xi, \theta_{\text{true}}) \in \mathcal{D}} \ell(\Phi(\xi, w), \theta_{\text{true}})$, with any loss metric $\ell$. Such a two-stage approach is efficient in terms of training the model, but it may lead to poor performance when a standard loss function is used. Performance can be improved if the loss function is carefully chosen to suit the task [11], but doing so is challenging for an arbitrary optimization problem.

Gradient-based end-to-end learning approaches in domains with optimization layers involved, e.g., decision-focused learning [43, 8], directly minimize Equation 1 as the training objective, which requires back-propagating through the optimization layer in Equation 2. This end-to-end approach is able to achieve better solution quality compared to two-stage learning, in principle. However, because the decision-focused approach has to repeatedly solve the optimization program and back-propagate through it, scalability becomes a serious issue. Additionally, the complex optimization layer can also jeopardize the smoothness of objective value, which is detrimental for training parameters of a neural network-based predictive model with gradient-based methods.

## 3 Surrogate Learning

The main idea of the surrogate approach is to replace Equation 2 with a carefully selected surrogate problem. To simplify Equation 2, we can linearly reparameterize $\mathbf{x} = P\mathbf{y}$, where $y \in \mathbb{R}^m$ with $m \ll n$ and $P \in \mathbb{R}^{n \times m}$,

$$\min_{\mathbf{y}} \quad g_P(\mathbf{y}, \theta) \coloneqq f(P\mathbf{y}, \theta) \quad \text{s.t.} \quad h(P\mathbf{y}) \leq 0, \quad AP\mathbf{y} = b \tag{3}$$

Since this reparameterization preserves all the equality and inequality constraints in Equation 2, we can easily transform a feasible low-dimensional solution $\mathbf{y}^*$ back to a feasible high-dimensional solution with $\mathbf{x}^* = P\mathbf{y}^*$. The low-dimensional surrogate is generally easier to solve, but lossy, because we restrict the feasible region to a hyperplane spanned by $P$. If we were to use a random reparameterization, the solution we recover from the surrogate problem could be far from the actual optimum in the original optimization problem, which could significantly degrade the solution quality.

This is why we need to learn the surrogate and its reparameterization matrix. Because we can differentiate through the surrogate optimization layer, we can estimate the impact of the reparameterization matrix on the final solution quality. This allows us to run gradient descent to learn the reparameterization matrix $P$. The process is shown in Figure 3. Notice that the surrogate problem also takes the prediction $\theta$ of the predictive model as input. This implies that we can jointly learn the predictive model and the reparameterization matrix by solely solving the cheaper surrogate problem.

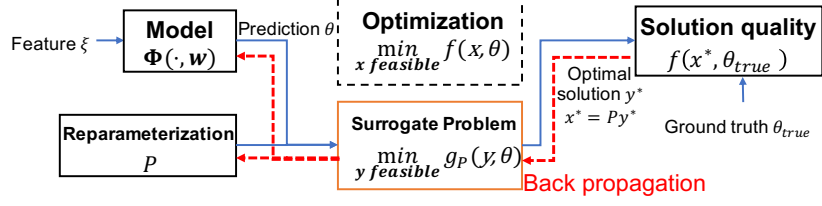

Figure 3: Surrogate decision-focused learning reparameterizes Equation 2 by $\mathbf{x} = P\mathbf{y}$ to get a surrogate model in Equation 3. Then, forward and backward passes go through the surrogate model with a lower dimensional input $\mathbf{y}$ to compute the optimal solution and train the model.

**Differentiable optimization** In order to differentiate through the optimization layer as shown in Figure 2, we can compute the derivative of the solution quality, evaluated on the optimal solution $\mathbf{x}^*$ and true parameter $\theta_{\text{true}}$, with respect to the model's weights $w$ by applying the chain rule:

$$\frac{df(\mathbf{x}^*, \theta_{\text{true}})}{dw} = \frac{df(\mathbf{x}^*, \theta_{\text{true}})}{d\mathbf{x}^*} \frac{d\mathbf{x}^*}{d\theta} \frac{d\theta}{dw}$$

where $\frac{d\mathbf{x}^*}{d\theta}$ can be obtained by differentiating through KKT conditions of the optimization problem.

Similarly, in Figure 3, we can apply the same technique to obtain the derivatives with respect to the weights $w$ and reparameterization matrix $P$:

$$\frac{df(\mathbf{x}^*, \theta_{\text{true}})}{dw} = \frac{df(\mathbf{x}^*, \theta_{\text{true}})}{d\mathbf{x}^*} \frac{d\mathbf{x}^*}{d\mathbf{y}^*} \frac{d\mathbf{y}^*}{d\theta} \frac{d\theta}{dw}, \quad \frac{df(\mathbf{x}^*, \theta_{\text{true}})}{dP} = \frac{df(\mathbf{x}^*, \theta_{\text{true}})}{d\mathbf{x}^*} \frac{d\mathbf{x}^*}{d\mathbf{y}^*} \frac{d\mathbf{y}^*}{dP}$$

where $\mathbf{y}^*$ is the optimal solution of the surrogate problem, $\mathbf{x}^* = P\mathbf{y}^*$, and $\frac{d\mathbf{y}^*}{dw}, \frac{d\mathbf{y}^*}{dP}$ can be computed by differentiating through the KKT conditions of the surrogate optimization problem.

# 4 Analysis of Linear Reparameterization

The following sections address three major theoretical aspects: (i) complexity of solving the surrogate problem, (ii) learning the reparameterization, and (iii) learning the predictive model.

## 4.1 Convexity and DR-Submodularity of the Reparameterized Problem

In this section, we assume the predictive model and the linear reparameterization are fixed. We prove below that convexity and continuous diminishing-return (DR) submodularity [6] of the original function $f$ is preserved after applying the reparameterization. This implies that the new surrogate problem can be efficiently solved by gradient descent or by Frank-Wolfe [7, 22, 16] with an approximation guarantee.

**Proposition 1.** *If $f$ is convex, then $g_P(\mathbf{y}, \theta) = f(P\mathbf{y}, \theta)$ is convex.*

**Proposition 2.** *If $f$ is DR-submodular and $P \geq 0$, then $g_P(\mathbf{y}, \theta) = f(P\mathbf{y}, \theta)$ is DR-submodular.*

## 4.2 Convexity of Reparameterization Learning

In this section, we assume the predictive model $\Phi$ is fixed. We want to analyze the convergence of learning the surrogate and its linear reparameterization $P$. Let us denote the optimal value of the optimization problem in the form of Equation 3 to be $\text{OPT}(\theta, P) := \min_{\mathbf{y} \text{ feasible}} g_P(\mathbf{y}, \theta) \in \mathbb{R}$. It would be ideal if $\text{OPT}(\theta, P)$ is convex in $P$ so that gradient descent would be guaranteed to recover the optimal reparameterization. Unfortunately, this is not true in general, despite the fact that we use a linear reparameterization: $\text{OPT}(\theta, P)$ is not even globally quasiconvex in $P$.

**Proposition 3.** *$OPT(\theta, P) = \min_{\mathbf{y} \text{ feasible}} g_P(\mathbf{y}, \theta)$ is not globally quasiconvex in $P$.*

Fortunately, we can guarantee the partial quasiconvexity of $\text{OPT}(\theta, P)$ in the following theorem:

**Theorem 1.** *If $f(\cdot, \theta)$ is quasiconvex, then $OPT(\theta, P) = \min_{\mathbf{y} \text{ feasible}} g_P(\mathbf{y}, \theta)$ is quasiconvex in $P_i$, the $i$-th column of matrix $P$, for any $1 \leq i \leq m$, where $P = [P_1, P_2, \ldots, P_m]$.*

This indicates that the problem of optimizing each meta-variable given the values of the others is tractable, providing at least some reason to think that the training landscape for the reparameterization is amenable to gradient descent. This theoretical motivation is complemented by our experiments, which show successful training with standard first-order methods.

### 4.3 Sample Complexity of Learning Predictive Model in Surrogate Problem

In this section, we fix the linear reparameterization and analyze the sample complexity of learning the predictive model to achieve small decision-focused loss in the objective value. We analyze a special case where our objective function $f$ is a linear function and the feasible region $S$ is compact, convex, and polyhedron. Given the hypothesis class of our model $\Phi \in \mathcal{H}$, we can use results from Balghiti et al. [10] to bound the Rademacher complexity and the generalization bound of the solution quality obtained from the surrogate problem. For any hypothesis class with a finite Natarajan dimension, the surrogate problem preserves the linearity of the objective function. Thus learning in the linear surrogate problem also *preserves the convergence of the generalization bound, and thus the convergence of the solution quality*. In the case of a linear hypothesis class $H = H_{\text{lin}}$, we can derive a closed-form bound. The Rademacher complexity depends on the dimensionality of the surrogate problem and the diameter of the feasible region, which can be shrunk by using a low-dimensional surrogate:

**Theorem 2.** *Let $\mathcal{H}_{lin}$ be the hypothesis class of all linear function mappings from $\xi \in \Xi \subset \mathbb{R}^p$ to $\theta \in \Theta \in \mathbb{R}^n$, and let $P \in \mathbb{R}^{n \times m}$ be a linear reparameterization used to construct the surrogate. The expected Rademacher complexity over $t$ i.i.d. random samples drawn from $\mathcal{D}$ can be bounded by:*

$$Rad^t(\mathcal{H}_{lin}) \leq 2mC\sqrt{\frac{2p\log(2mt\,\|P^+\|\,\rho_2(S))}{t}} + O(\frac{1}{t}) \tag{4}$$

*where $C := sup_\theta(max_x f(x,\theta) - min_{x'} f(x',\theta))$ is the gap between the optimal solution quality and the worst solution quality, $\rho_2(S)$ is the diameter of the set $S$, and $P^+$ is the pseudoinverse.*

Equation 4 gives a bound on the Rademacher complexity, an upper bound on the generalization error with $t$ samples given. Although a lower dimensional surrogate leads to less representational power (i.e., lower decision quality), it also leads to better generalizability. This implies that we have to choose an appropriate reparameterization size to balance representational power and generalizability.

## 5 Experiments

We conduct experiments on three different domains where decision-focused learning has been applied: (i) adversarial behavior learning in network security games with a non-convex objective [41], (ii) movie recommendation with a submodular objective [43], and (iii) portfolio optimization problem with a convex quadratic objective [13]. Throughout all the experiments, we compare the performance and the scalability of the surrogate learning (**surrogate**), two-stage (**TS**), and decision-focused (**DF**) learning approaches. Performance is measured in terms of regret, which is defined as the difference between the achieved solution quality and the solution quality if the unobserved parameters $\theta^*$ were observed directly—smaller is better. To compare scalability, we show the training time per epoch and inference time. The inference time corresponds to the time required to compute a decision for all instances in the testing set after training is finished. A short inference time may have intrinsic value, e.g., allowing the application to be run in edge computing settings. All methods are trained using gradient descent with optimizer Adam [25] with learning rate 0.01 and repeated over 30 independent runs to get the average. Each model is trained for at most 100 epochs with early stopping [34] criteria when 3 consecutive non-improving epochs occur on the validation set. The reparameterization size is set to be 10% of the problem size throughout all three examples[1].

### 5.1 Adversarial Behavior Learning and Interdiction Games

Given a network structure $G = (V, E)$, a NSG (network security game) [42, 14, 38] models the interaction between the defender, who places checkpoints on a limited number of edges in the graph,

and an attacker who attempts to travel from a source to any of a set of target nodes in order to maximize the expected reward. The NSG is an extension of Stackelberg security games [39, 23], meaning that the defender commits to a mixed strategy first, after which the attacker chooses the path (having observed the defender's mixed strategy but not the sampled pure strategy). In practice, the attacker is not perfectly rational. Instead, the defender can attempt to predict the attacker's boundedly rational choice of path by using the known features of the nodes en route (e.g., accessibility or safety of hops) together with previous examples of paths chosen by the attacker.

Once the parameters $\theta$ of the attacker behavioral model are given, finding the optimal defender's strategy reduces to an optimization problem $\max f(\mathbf{x}, \theta)$ where $\mathbf{x}_e$ is the probability of covering edge $e \in E$ and $f$ gives the defender's expected utility for playing mixed strategy $\mathbf{x}$ when the attacker's response is determined by $\theta$. The defender must also satisfy the budget constraint $\sum_{e \in E} \mathbf{x}_e \leq k$ where $k = 3$ is the total defender resources. We use a GCN (graph convolutional network) [31, 26, 18] to represent the predictive model of the attacker. We assume the attacker follows reactive Markovian behavior [41, 17], meaning that the attacker follows a random walk through the graph, where the probability of transitioning across a given edge $(u, v)$ is a function of the defender's strategy $\mathbf{x}$ and an unknown parameter $\theta_v$ representing the "attractiveness" of node $v$. The walk stops when the attacker either is intercepted by crossing an edge covered by the defender or reaches a target. The defender's utility is $-u(t)$ if the attacker reaches target $t$ and 0 otherwise, and $f$ takes an expectation over both the random placement of the defender's checkpoints (determined by $\mathbf{x}$) and the attacker's random walk (determined by $\mathbf{x}$ and $\theta$). Our goal is to learn a GCN which takes node features as input and outputs the attractiveness over nodes $\theta$.

**Experimental setup:** We generate random geometric graphs of varying sizes with radius 0.2 in a unit square. We select 5 nodes uniformly at random as targets with payoffs $u(t) \sim \text{Unif}(5, 10)$ and 5 nodes as sources where the attacker chooses uniformly at random from. The ground truth attractiveness value $\theta_v$ of node $v \in V$ is proportional to the proximity to the closest target plus a random perturbation sampled as $\text{Unif}(-1, 1)$ which models idiosyncrasies in the attacker's preferences. The node features $\xi$ are generated as $\xi = \text{GCN}(\theta) + 0.2\mathcal{N}(0, 1)$, where GCN is a randomly initialized GCN with four convolutional layers and three fully connected layers. This generates random features with correlations between $\xi_v$ (the features of node $v$) and both $\theta_v$ and the features of nearby nodes. Such correlation is expected for real networks where neighboring locations are likely to be similar. The defender's predictive model (distinct from the generative model) uses two convolutional and two fully connected layers, modeling a scenario where the true generative process is more complex than the learned model. We generate 35 random $(\xi, \theta)$ pairs for the training set, 5 for validation, and 10 for testing. Since decision-focused (DF) learning fails to scale up to larger instances, we additionally compare to a block-variable sampling approach specialized to NSG [41] (**block**), which can speed up the backward pass by back-propagating through randomly sampled variables.

### 5.2 Movie Recommendation and Broadcasting Problem

In this domain, a broadcasting company chooses $k$ movies out of a set of $n$ available to acquire and show to their customers $C$. $k$ reflects a budget constraint. Each user watches their favorite $T$ movies, with a linear valuation for the movies they watch. This is a variant of the classic facility location problem; similar domains have been used to benchmark submodular optimization algorithms [28, 9]. In our case, the additional complication is that the user's preferences are unknown. Instead, the company uses user's past behavior to predict $\theta_{ij} \in [0, 1]$, the preference score of user $j$ for movie $i$.

The company would like to maximize the overall satisfaction of users without exceeding the budget constraint $k = 10$. $\{\mathbf{x}_i\}_{i \in \{1,2,\dots,n\}}$ represents the decision of whether to acquire movie $i$ or not. Once the preferences $\theta_{ij}$ are given, the company wants to maximize the objective function:

$$f(\mathbf{x}) := \sum_{j \in C} \text{user } j\text{'s satisfaction} = \sum_{j \in C} \max_{\substack{z_j \in \{0,1\}^n \\ \text{s.t.} \sum_i z_{ij} = T}} \sum_{i \in \{1,2,\dots,n\}} x_i z_{ij} \theta_{ij} \quad (5)$$

where $z_j$ denotes the user $j$'s selection over movies.

**Experimental setup:** We use neural collaborative filtering [20] to learn the user preferences. Commonly used in recommendation systems, the idea is to learn an embedding for each movie and

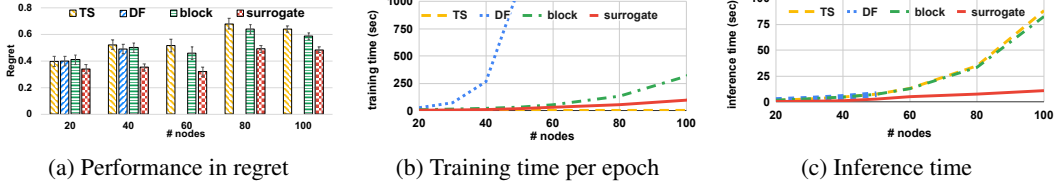

(a) Performance in regret     (b) Training time per epoch     (c) Inference time

Figure 4: Experimental results in network security games with a non-convex optimization problem.

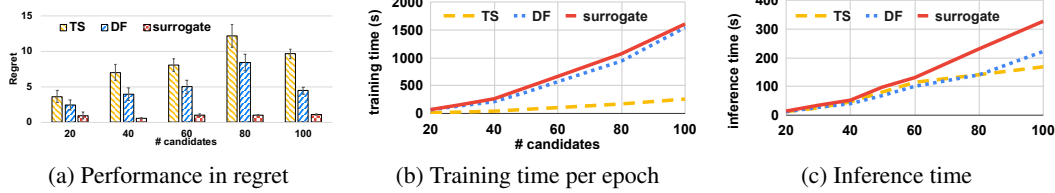

(a) Performance in regret     (b) Training time per epoch     (c) Inference time

Figure 5: Experimental results in movie recommendation with a submodular objective. Surrogate achieves much better performance by smoothing the training landscape.

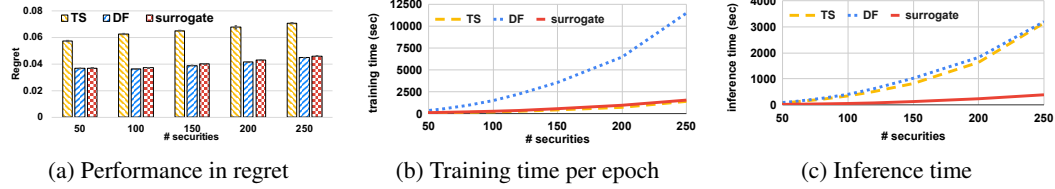

(a) Performance in regret     (b) Training time per epoch     (c) Inference time

Figure 6: Experimental results in portfolio optimization with a convex optimization problem. Surrogate performs comparably, but achieves a 7-fold speedup in training and inference.

user. The ratings are computed by feeding the concatenated user's and movie's embeddings to a neural network with fully connected layers. We use MovieLens [19] as our dataset. The MovieLens dataset includes 25M ratings over 62,000 movies by 162,000 users. We first randomly select $n$ movies as our broadcasting candidates. We additionally select 200 movies and use the users' ratings on the movies as the users' features. Then we split the users into disjoint groups of size 100 and each group serves as an instance of broadcasting task, where we want to choose $k = 10$ from the $n$ candidate movies to recommend to the group members. Each user chooses $T = 3$ movies. 70% of the user groups are used for training, 10% for validation, and 20% for testing.

### 5.3 Stock Market Portfolio Optimization

Portfolio optimization can be treated as an optimization problem with missing parameters [33], where the return and the covariance between stocks in the next time step are not known in advance. We learn a model that takes features for each security and outputs the predicted future return. We adopt the classic Markowitz [29, 30] problem setup, where investors are risk-averse and wish to maximize a weighted sum of the immediate net profit and the risk penalty. The investor chooses a vector $\mathbf{x} \geq 0$ with $\sum \mathbf{x}_i = 1$, where $\mathbf{x}_i$ represents the fraction of money invested in security $i$. The investor aims to maximize the penalized immediate return $f(\mathbf{x}) \coloneqq p^\top \mathbf{x} - \lambda \mathbf{x}^\top Q \mathbf{x}$, where $p$ is the immediate net return of all securities and $Q \in \mathbb{R}^{n \times n}$ is a positive semidefinite matrix representing the covariance between the returns of different securities. A high covariance implies two securities are highly correlated and thus it is more risky to invest in both. We set the risk aversion constant to be $\lambda = 2$.

**Experimental setup:** We use historical daily price and volume data from 2004 to 2017 downloaded from the Quandl WIKI dataset [35]. We evaluate on the SP500, a collection of the 505 largest companies representing the American market. Our goal is to generate daily portfolios of stocks from a given set of candidates. Ground truth returns are computed from the time series of prices, while the ground truth covariance of two securities at a given time step is set to be the cosine similarity of their returns in the next 10 time steps. We take the previous prices and rolling averages at a given time step as features to predict the returns for the next time step. We learn the immediate return $p$ via a neural network with two fully connected layers with 100 nodes each. To predict the covariance

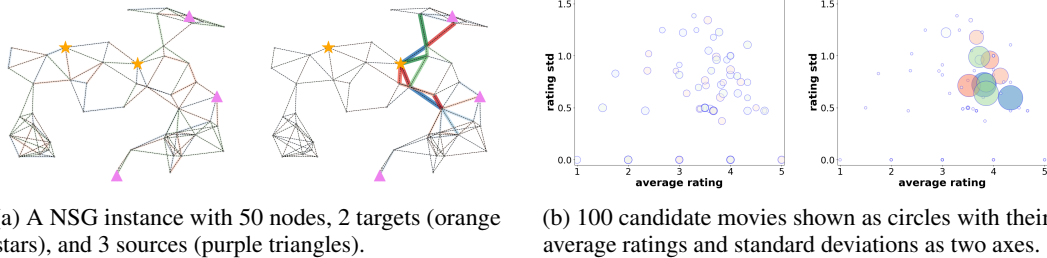

(a) A NSG instance with 50 nodes, 2 targets (orange stars), and 3 sources (purple triangles).

(b) 100 candidate movies shown as circles with their average ratings and standard deviations as two axes.

Figure 7: These plots visualize how the surrogate captures the underlying problem structure. Both domains use a reparameterization with 3 meta-variables, each shown in red, blue, and green. The color indicates the most significant meta-variable governing the edge or circle, while the color intensity and size represent the weights put on it. The left figure in both domains shows the initial reparameterization, while the right figure shows the reparameterization after 20 epochs of training.

matrix $Q$, we learn an 32-dimensional embedding for each security, and the predicted covariance between two securities is the cosine similarity of their embeddings. We chronologically split the dataset into training, validation, and test sets with 70%, 10%, and 20% of the data respectively.

## 6   Discussion of Experimental Results

**Performance:**    Figures 4(a), 5(a), and 6(a) compare the regret of our surrogate approach to the other approaches. In the non-convex (Figure 4(a)) and submodular (Figure 5(a)) settings, we see a larger improvement in solution quality relative to decision-focused learning. This is due to the huge number of local minima and plateaus in these two settings where two-stage and decision-focused approaches can get stuck. For example, when an incorrect prediction is given in the movie recommendation domain, some recommended movies could have no users watching them, resulting in a sparse gradient due to non-smoothness induced by the max in the objective function. The surrogate approach can instead spread the sparse gradient by binding variables with meta-variables, alleviating gradient sparsity. We see relatively less performance improvement (compared to decision-focused) when the optimization problem is strongly convex and hence smoother (Figure 6(a)), though the surrogate approach still achieves similar performance to the decision-focused approach.

**Scalability:**    When the objective function is non-convex (Figure 4(b), 4(c)), our surrogate approach yields substantially faster training than standard decision-focused learning approaches (DF and block). The boost is due to the dimensionality reduction of the surrogate optimization problem, which can lead to speedups in solving the surrogate problem and back-propagating through the KKT conditions. While the two-stage approach avoids solving the optimization problem in the training phase (trading off solution quality), at test time, it still has to solve the expensive optimization problem, resulting a similarly expensive inference runtime in Figure 4(c).

When the objective function is submodular (Figure 5(b), 5(c)), the blackbox optimization solver [40] we use in all experiments converges very quickly for the decision-focused method, resulting in training times comparable to our surrogate approach. However, Figure 5(a) shows that the decision-focused approach converges to a solution with very poor quality, indicating that rapid convergence may be a symptom of the uninformative local minima that the decision-focused method becomes trapped in.

Lastly, when the optimization problem is a quadratic program (Figure 6(b), 6(c)), solving the optimization problem can take cubic time, resulting in around a cubic speedup from the dimensionality reduction offered by our surrogate. Consequently, we see 7-fold faster training and inference times.

**Visualization:**    We visualize the reparameterization for the NSG and movie recommendation domains in Figure 7. The initial reparameterization is shown in Figure 7(a) and 7(b). Initially, the weights put on the meta-variables are randomly chosen and no significant problem structure—no edge or circle colors—can be seen. After 20 epochs of training, in Figure 7(a), the surrogate starts putting emphasis on some important cuts between the sources and the targets, and in Figure 7(b), the surrogate is focused on distinguishing between different top-rated movies with some variance in opinions to specialize the recommendation task. Interestingly, in Figure 7(b), the surrogate puts

less weight on movies with high average rating but low standard deviation because these movies are very likely undersampled and we do not have enough people watching them in our training data. Overall, adaptively adjusting the surrogate model allows us to extract the underlying structure of the optimization problem using few meta-variables. These visualizations also help us understand how focuses are shifted between different decision variables.

# 7  Conclusion

In this paper, we focus on the shortcomings of scalability and solution quality that arise in end-to-end decision-focused learning due to the introduction of the differentiable optimization layer. We address these two shortcomings by learning a compact surrogate, with a learnable linear reparameterization matrix, to substitute for the expensive optimization layer. This surrogate can be jointly trained with the predictive model by back-propagating through the surrogate layer. Theoretically, we analyze the complexity of the induced surrogate problem and the complexity of learning the surrogate and the predictive model. Empirically, we show this surrogate learning approach leads to improvement in scalability and solution quality in three domains: a non-convex adversarial modeling problem, a submodular recommendation problem, and a convex portfolio optimization problem.

## Broader impact:

End-to-end approaches can perform better in data-poor settings, improving access to the benefits of machine learning systems for communities that are resource constrained. Standard two-stage approaches typically requires enough data to learn well across the data distribution. In many domains focused on social impact such as wildlife conservation, limited data can be collected and the resources are also very limited. End-to-end learning is usually more favorable than two-stage approach under these circumstances; it can achieve higher quality results despite data limitations compared to two-stage approaches. This paper reduces the computational costs of end-to-end learning and increases the performance benefits.

But such performance improvements may come with a cost in transferability because the end-to-end learning task is specialized towards particular decisions, whereas a prediction-only model from the two-stage predict-then-optimize framework might be used for different decision making tasks in the same domain. Thus, the predictive model trained for a particular decision-making task in the end-to-end framework is not necessarily as interpretable or transferable as a model trained for prediction only. For real-world tasks, there would need to be careful analysis of cost-benefit of applying an end-to-end approach vis-a-vis a two-stage approach particularly if issues of interpretability and transferrability are critical; in some domains these may be crucial. Further research is required to improve upon these issues in the end-to-end learning approach.

## Acknowledgement

This research was supported by MURI Grant Number W911NF-17-1-0370 and W911NF-18-1-0208. The computations in this paper were run on the FASRC Cannon cluster supported by the FAS Division of Science Research Computing Group at Harvard University.

## Footnotes

[1]The implementation of this paper can be found in the following link: `https://github.com/guaguakai/surrogate-optimization-learning`

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
