[Supplementary Material]

**Appendix**

 **9 Preservation of Convexity and Submodularity**

 **Proposition 1.** *If $f$ is convex, then $g_P(\mathbf{y}, \theta) = f(P\mathbf{y}, \theta)$ is convex.*

 *Proof.* The convexity can be simply verified by computing the second-order derivative:

$$\frac{d^2 g}{d\mathbf{y}^2} = \frac{d^2 f(P\mathbf{y}, \theta)}{d\mathbf{y}^2} = P^\top \frac{d^2 f}{d\mathbf{x}^2} P \succeq 0$$

 where the last inequality comes from the convexity of $f$, i.e., $\frac{d^2 f}{d\mathbf{x}^2} \succeq 0$.  □

 **Proposition 2.** *If $f$ is DR-submodular and $P \geq 0$, then $g_P(\mathbf{y}, \theta) = f(P\mathbf{y}, \theta)$ is DR-submodular.*

 *Proof.* Assume $f$ has the property of diminishing return submodularity (DR-submodular) [7]. Ac-
 cording to definition of continuous DR-submodularity, we have:

$$\nabla^2_{\mathbf{x}_i, \mathbf{x}_j} f(\mathbf{x}, \theta) \leq 0 \; \forall i, j \in [n], \mathbf{y}$$

 After applying the reparameterization, we can write:

$$g_P(\mathbf{y}, \theta) = f(\mathbf{x}, \theta)$$

 and the second-order derivative:

$$\nabla^2_{\mathbf{y}} g_P(\mathbf{y}, \theta) = P^\top \nabla^2_{\mathbf{x}} f_P(\mathbf{x}, \theta) P \leq 0$$

 Since all the entries of $P$ are non-negative and all the entries of $\nabla^2_{\mathbf{x}} f_P(\mathbf{x}, \theta)$ are non-positive by
 DR-submodularity, the product $\nabla^2_{\mathbf{y}} g_P(\mathbf{y}, \theta)$ also has all the entries being non-positive, which satisfies
 the definition of DR-submodularity.  □

 **10 Quasiconvexity in Reparameterization Matrix**

 **Proposition 3.** $OPT(\theta, P) = \min_{\mathbf{y} \; feasible} g_P(\mathbf{y}, \theta)$ *is not globally quasiconvex in $P$.*

 *Proof.* Without loss of generality, let us ignore the effect of $\theta$ and write $g_P(\mathbf{y}) = f(P\mathbf{x})$. In this
 proof, we will construct a strongly convex function $f$ where the induced optimal value function
 $\text{OPT}(P) := \min_{\mathbf{y}} g_P(\mathbf{y})$ is not quasiconvex.

 Consider $\mathbf{x} = [\mathbf{x}_1, \mathbf{x}_2, \mathbf{x}_3]^\top \in \mathbb{R}^3$. Define $f(\mathbf{x}) = \left\| \mathbf{x} - \begin{pmatrix} 1 \\ 1 \\ 1 \end{pmatrix} \right\|^2 \geq 0$ for all $\mathbf{x} \in \mathbb{R}^3$. Define $P =$

 $\begin{pmatrix} 1 & 0 \\ 1 & 0 \\ 0 & 2 \end{pmatrix}$ and $P' = \begin{pmatrix} 0 & 1 \\ 0 & 1 \\ 2 & 0 \end{pmatrix}$. Apparently, $\mathbf{x}^* = \begin{pmatrix} 1 \\ 1 \\ 1 \end{pmatrix} = P \begin{pmatrix} 1 \\ 0.5 \end{pmatrix}$ and $\mathbf{x}^* = \begin{pmatrix} 1 \\ 1 \\ 1 \end{pmatrix} = P' \begin{pmatrix} 0.5 \\ 1 \end{pmatrix}$ are

 both achievable. So the optimal values $\text{OPT}(P) = \text{OPT}(P') = 0$. But the combination $P'' = \frac{1}{2} P +$

 $\frac{1}{2} P' = \begin{pmatrix} 0.5 & 0.5 \\ 0.5 & 0.5 \\ 1 & 1 \end{pmatrix}$ cannot, which results in an optimal value $OPT(P'') = \min_{\mathbf{y}} g_{P''}(\mathbf{y}) => 0$

 since $\begin{pmatrix} 1 \\ 1 \\ 1 \end{pmatrix} \notin \text{span}(P'')$. This implies $\text{OPT}(\frac{1}{2} P + \frac{1}{2} P') = \text{OPT}(P'') > 0 = \frac{1}{2}\text{OPT}(P) + \frac{1}{2}\text{OPT}(P')$.

 Thus $\text{OPT}(P)$ is not globally convex in the feasible domain.  □

 **Theorem 1.** *If $f(\cdot, \theta)$ is quasiconvex, then $OPT(\theta, P) = \min_{\mathbf{y} \; feasible} g_P(\mathbf{y}, \theta)$ is quasiconvex in $P_i$*
 *for any $1 \leq i \leq m$, where $P = [P_1, P_2, \ldots, P_m] \geq 0$.*

*Proof.* Let us assume $P = [p_1, p_2, ..., p_m]$ and $P' = [p'_1, p'_2, ..., p'_m]$, where $p_i = p'_i \ \forall i \neq 1$ with only the first column different. In the optimization problem parameterized by $P$, there is an optimal solution $x = \sum_{i=1}^{m} p_i y_i$, $y_i \geq 0 \ \forall i$. Similarly, there is an optimal solution $x' = \sum_{i=1}^{m} p'_i y'_i$, $y'_i \geq 0 \ \forall i$ for the optimization problem parameterized by $P'$. We know that $f(x) = h(P)$, $f(x') = h(P')$. Denote $P'' = cP + (1-c)P' = [p''_1, p''_2, ..., p''_m]$ to be a convex combination of $P$ and $P'$. Clearly, $p''_1 = cp_1 + (1-c)p'_1$ and $p''_i = p_i = p'_i \ \forall i \neq 1$. Then we can construct a solution

$$
\begin{aligned}
x'' &= \frac{1}{\frac{c}{y_1} + \frac{1-c}{y'_1}} \left( \frac{c}{y_1} x + \frac{1-c}{y'_1} x' \right) \\
&= \frac{1}{\frac{c}{y_1} + \frac{1-c}{y'_1}} \left( \frac{c}{y_1} \sum_{i=1}^{m} p_i y_i + \frac{1-c}{y'_1} \sum_{i=1}^{m} p'_i y'_i \right) \\
&= \frac{1}{\frac{c}{y_1} + \frac{1-c}{y'_1}} (cp_1 + (1-c)p'_1) + \frac{1}{\frac{c}{y_1} + \frac{1-c}{y'_1}} \sum_{i=2}^{m} p_i \left( \frac{y_i}{y_1} + \frac{y'_i}{y'_1} \right) \\
&\in \text{Span}(P'')
\end{aligned}
$$

Thus, $x''$ is a feasible solution in the optimization problem parameterized by $P''$. By the convexity of $f$, we also know that

$$
\begin{aligned}
h(cP + (1-c)P') = h(P'') &\leq f(x'') \\
&= f\left( \frac{1}{\frac{c}{y_1} + \frac{1-c}{y'_1}} \left( \frac{c}{y_1} x + \frac{1-c}{y'_1} x' \right) \right) \\
&\leq \max(f(x), f(x')) \\
&= \max(h(P), h(P'))
\end{aligned}
$$

When one of $y_1, y'_1$ is 0, without loss of generality we assume $y_1 = 0$. Then we can construct a solution $x'' = x$ which is still feasible in the optimization problem parameterized by $P'' = cP + (1-c)P'$. Then we have the following:

$$
h(P'') \leq f(x'') = f(x) = h(P) \leq \max(h(P), h(P'))
$$

which concludes the proof. $\square$

# 11 Sample Complexity of Learning Predictive Model in Surrogate Problem

**Theorem 2.** *Let $\mathcal{H}_{lin}$ be the hypothesis class of all linear function mappings from $\xi \in \Xi \subset \mathbb{R}^p$ to $\theta \in \Theta \in \mathbb{R}^n$, and let $P \in \mathbb{R}^{n \times m}$ be a linear reparameterization used to construct the surrogate. The expected Rademacher complexity over $t$ i.i.d. random samples drawn from $\mathcal{D}$ can be bounded by:*

$$
Rad^t(\mathcal{H}_{lin}) \leq 2mC \sqrt{\frac{2p \log(2mt \, \|P^+\| \, \rho_2(S))}{t}} + O(\frac{1}{t}) \tag{4}
$$

*where $C$ is the gap between the optimal solution quality and the worst solution quality, $\rho_2(S)$ is the diameter of the set $S$, and $P^+$ is the pseudoinverse.*

The proof of Theorem 2 relies on the results given by Balghiti et al. [11]. Balghiti et al. analyzed the sample complexity of predict-then-optimize framework when the optimization problem is a constrained linear optimization problem.

The sample complexity depends on the hypothesis class $\mathcal{H}$, mapping from the feature space $\Xi$ to the parameter space $\Theta$. $\mathbf{x}^*_S(\theta) = \operatorname{argmin}_{\mathbf{x} \in S} f(\mathbf{x}, \theta)$ characterizes the optimal solution with given parameter $\theta \in \Theta$ and feasible region $S$. This can be obtained by solving any linear program solver with given parameters $\theta$. The optimization gap with given parameter $P$ is defined as $\omega_S(\theta) :=$ $\max_{\mathbf{x} \in S} f(\mathbf{x}, \theta) - \min_{\mathbf{x} \in S} f(\mathbf{x}, \theta)$, and $\omega_S(\Theta) := \sup_{\theta \in \Theta} \omega_S(\theta)$ is defined as the upper bound on optimization gap of all the possible parameter $\theta \in \Theta$. $\mathbf{x}^*(\mathcal{H}) := \{\xi \to \mathbf{x}^*(\Phi(\xi)) | \Phi \in \mathcal{H}\}$ is the set of all function mappings from features $\xi$ to the predictive parameters $\theta = \Phi(\xi)$ and then to the optimal solution $\mathbf{x}^*(\theta)$.

**Definition 1** (Natarajan dimension). *Suppose that $S$ is a polyhedron and $\mathfrak{S}$ is the set of its extreme points. Let $\mathcal{F} \in \mathfrak{S}^{\Xi}$ be a hypothesis space of function mappings from $\Xi$ to $\mathfrak{S}$, and let $A \in \Xi$ to be given. We say that $\mathcal{F}$ shatters $A$ if there exists $g_1, g_2 \in \mathcal{F}$ such that*

- $g_1(\xi) \neq g_2(\xi) \; \forall \xi \in A$.

- *For all $B \subset A$, there exists $g \in \mathcal{F}$ such that (i) for all $\xi \in B$, $g(\xi) = g_1(\xi)$ and (ii) for all $\xi \in A \backslash B$, $g(\xi) = g_2(\xi)$.*

*The Natarajan dimension of $\mathcal{F}$, denoted by $d_N(\mathcal{F})$, is the maximum cardinality of a set N-shattered by $\mathcal{F}$.*

We first state their results below:

**Theorem 3** (Balghiti et al. [11] Theorem 2). *Suppose that $S$ is a polyhedron and $\mathfrak{S}$ is the set of its extreme points. Let $\mathcal{H}$ be a family of functions mapping from features $\Xi$ to parameters $\Theta \in \mathbb{R}^n$ with decision variable $\mathbf{x} \in \mathbb{R}^n$ and objective function $f(\mathbf{x}, \theta) = \theta^\top \mathbf{x}$. Then we have that*

$$Rad^t(\mathcal{H}) \leq \omega_S^*(\Theta) \sqrt{\frac{2 d_N(\mathbf{x}^*(\mathcal{H})) \log(t|\mathfrak{S}|^2)}{t}}. \tag{5}$$

*where $Rad^t$ denotes the Radamacher complexity averaging over all the possible realization of $t$ i.i.d. samples drawn from distribution $\mathcal{D}$.*

The following corollary provided by Balghiti et al. [11] introduces a bound on Natarajan dimension of linear hypothesis class $\mathcal{H}$, mapping from $\Xi \in \mathbb{R}^p$ to $\Theta \in \mathbb{R}^n$:

**Corollary 1** (Balghiti et al. [11] Corollary 1). *Suppose that $S$ is a polyhedron and $\mathfrak{S}$ is the set of its extreme points. Let $\mathcal{H}_{lin}$ be the hypothesis class of all linear functions, i.e., $\mathcal{H}_{lin} = \{\xi \to B\xi | B \in \mathbb{R}^{n \times p}\}$. Then we have*

$$d_N(\mathbf{x}^*(\mathcal{H}_{lin})) \leq np \tag{6}$$

Also $|\mathfrak{S}|$ can be estimated by constructing an $\epsilon$-covering of the feasible region by open balls with radius $\epsilon$. Let $\hat{\mathfrak{S}}_\epsilon$ be the centers of all these open balls. We can choose $\epsilon = \frac{1}{t}$ and the number of open balls required to cover $S$ can be estimated by

$$|\hat{\mathfrak{S}}_\epsilon| \leq \left(2 t \rho_2(S) \sqrt{n}\right)^n \tag{7}$$

Combining Equation 5, 6, and 7, the Radamacher complexity can be bounded by:

**Corollary 2** (Balghiti et al. [11] Corollary 2).

$$Rad^t(\mathcal{H}_{lin}) \leq 2n\omega_S(\Theta) \sqrt{\frac{2p \log(2nt\rho_2(S))}{t}} + O(\frac{1}{t}) \tag{8}$$

Now we are ready to prove Theorem 2:

*Proof of Theorem 2.* Now let us consider our case. We have a linear mapping from features $\xi \in Xi \subset \mathbb{R}^p$ to the parameters $\theta = B\xi \in \Theta \in \mathbb{R}^n$ with $B \in \mathbb{R}^{n \times p}$. The objective function is formed by

$$g_P(\mathbf{y}, \theta) = f(P\mathbf{y}, \theta) = \theta^\top P\mathbf{y} = (P^\top \theta)^\top \mathbf{y} = (P^\top B\xi)^\top \mathbf{y} \tag{9}$$

This is equivalent to have a linear mapping from $\xi \in \Xi \subset \mathbb{R}^p$ to $\theta' = P^\top B\xi$ where $P^\top B \in \mathbb{R}^{m \times p}$, and the objective function is just $g_P(\mathbf{y}, \theta') = \theta'^\top \mathbf{y}$. This yields a similar bound but with a smaller dimension $m \ll n$ as in Equation 10:

$$]Rad^t(\mathcal{H}_{lin}) \leq 2m\omega_S(\Theta) \sqrt{\frac{2p \log(2mt\rho_2(S'))}{t}} + O(\frac{1}{t}) \tag{10}$$

where $\omega_S(\Theta)$ is unchanged because the optimality gap is not changed by the reparameterization. The only thing changed except for the substitution of $m$ is that the feasible region $S'$ is now defined in a

527 lower-dimensional space under reparameterization $P$. But since $\forall \mathbf{y} \in S'$, we have $P\mathbf{y} \in S$ too. So
528 the diameter of the new feasible region can also be bounded by:

$$\begin{aligned}
\rho(S') &= \max_{\mathbf{y}, \mathbf{y}' \in S'} \|\mathbf{y} - \mathbf{y}'\| \\
&= \max_{\mathbf{y}, \mathbf{y}' \in S'} \|P^+ P(\mathbf{y} - \mathbf{y}')\| \\
&= \max_{\mathbf{y}, \mathbf{y}' \in S'} \|P^+ (P\mathbf{y} - P\mathbf{y}')\| \\
&\leq \max_{\mathbf{x}, \mathbf{x}' \in S'} \|P^+ (\mathbf{x} - \mathbf{x}')\| \\
&\leq \|P^+\| \max_{\mathbf{x}, \mathbf{x}' \in S'} \|\mathbf{x} - \mathbf{x}'\| \\
&= \|P^+\| \rho(S)
\end{aligned}$$

529 where $P^+ \in \mathbb{R}^{m \times n}$ is the pseudoinverse of the reparameterization matrix $P$ with $P^+ P = I \in \mathbb{R}^{m \times m}$
530 (assuming the matrix does not collapse). Substituting the term $\rho(S')$ in Equation 10, we can get the
531 bound on the Radamacher complexity in Equation 4, which concludes the proof of Theorem 2. $\qquad\square$

## 12 Non-linear Reparameterization

533 The main reason that we use a linear reparameterization is to maintain the convexity of the inequality
534 constraints and the linearity of the equality constraints. Instead, if we apply a convex reparameteriza-
535 tion $\mathbf{x} = P(\mathbf{y})$, e.g., an input convex neural network [3], then the inequality constraints will remain
536 convex but the equality constraints will no longer be affine anymore. So such convex reparameter-
537 ization can be useful when there is no equality constraint. Lastly, we can still apply non-convex
538 reparameterization but it can create non-convex inequality and equality constraints, which can be
539 challenging to solve. All of these imply that the choice of reparameterization should depend on the
540 type of optimization problem to make sure we do not lose the scalability while solving the surrogate
541 problem.

## 13 Computing Infrastructure

543 All experiments were run on the computing cluster, where each node configured with 2 Intel Xeon
544 Cascade Lake CPUs, 184 GB of RAM, and 70 GB of local scratch space. Within each experiment,
545 we did not implement parallelization. So each experiment was purely run on a single CPU core. The
546 main bottleneck of the computation is on solving the optimization problem, where we use Scipy [41]
547 blackbox optimization solver. No GPU was used to train the neural network and throughout the
548 experiments.