[Reviews · NeurIPS 2020]

Review 1

Summary and Contributions: This paper presents a surrogate based end-to-end method to solve optimization problems where important decision parameters are unknown (and hence need to be inferred). In order to leverage the usual scalability and smoothness issues caused by usual “predict and optimize” approaches, the optimization problem is reparameterized by a learned surrogate that allows optimizing in a smaller dimensional space through a linear transformation. The authors provide some theoretical guarantees on the convexity and submodularity properties of the transformed optimization problem, on the partial pseudo-convexity of the re-parameterized learning problem and on the sample complexity of the predictive model learning task, thus arguing that such an approach should be in practice tractable and learnable with gradient descent (in spite of the non-convexity of the reparameterized learning problem). They also provide experiments carried out in three types of configurations, including convex, non-convex and submodular optimization problems and compare both the performance and scalability of the approach to (1) a two-stage approach that uses a separate ground truth-based model prior to optimization on the inferred parameters and (2) a classical end-to-end decision-focused method. The presented approach seems to significantly improve the performances when solving problems with numerous possible local optima (i.e. non-convex and submodular cases). Moreover significant gains in terms of training and inference times appear to be made in convex cases and in terms of inference time in non-convex cases thanks to the dimension reduction induced by the change of parameterization.

Strengths: Some theoretical guarantees are given and used to justify, alongside with the experimental results, the feasibility of training and the sample complexity of the method in diverse types of problems. Experiments seem to efficiently include most of the useful configurations to assess the approach properties, both in terms of optimization problem properties (convexity and submodularity), regarding influence of the proposed modifications in comparison to existing approaches and regarding computational costs versus “raw” performances. The results seem compelling in terms of raw performances in the case of non-convex and submodular optimization problems and in terms of computational cost in non-submodular problems. The proposed method is fairly intuitive and natural: the linear reparametrization of the optimization problem seems to be a rather cheap and easily justified improvement which appears to lead to significant improvements in the cases mentioned above.

Weaknesses: The problem being solved seems to be quite restricted : the paper focuses on the mentioned “predict-then-optimize”, thus does not give much insight of the other types of approaches (perhaps less “model based”) framed in a similar context of finding optimal values of an objective with the help of informative covariates. At least a larger description of the landscape of methods used would have helped to get the true and possibly larger impact of this work and to replace it in a more general context.

Correctness: The empirical methodology appears to be correct and outlines efficiently the properties of the method compared to the two other baselines. As noted previously a comparison with other classes of methods attempting to solve similar problems would have been interesting to evaluate the performances of the approach.

Clarity: The paper is clear, concise and globally well written. The ideas and the main justifications are clearly described. Some “typos” are to notice (lines 126, 130 ...), and some of them could harm the clarity of the content (exemple : y = Py in the caption of figure 3).

Relation to Prior Work: The main framework and problem tackled in this work appear to be fairly specific in its very formulation. A wider review of different approaches (e.g. outside the “predict and optimize” framework) tackling similar problems of optimizing an objective from covariates would have been good to place this work in more general context and to outline its precise contributions and impact, as stated previously.

Reproducibility: Yes

Additional Feedback: Items to check and fix: - Caption of Fig. 3: **x** = Py - Typo in title of Sec. 4: Analysis of Linear **Reparameterization** (line 126) - Line 130: The sentence starting with “In this section, …” is incomplete. - Equation in the middle of Sec. 5.2 is not easy to understand, the mix of different sizes in the sums and such make it quite tricky. Moreover, such equation deserves to be numbered. ** Post-rebuttal: I have read the rebuttal and maintain my evaluation, this is a good submission. **


Review 2

Summary and Contributions: In the context of optimization in uncertain environment (i.e., some parameters of the optimization problem itself are unknown and depend on the solution itself), end-to-end learning of both the mapping from solution to unknown parameters and the optimiation problem has been made possible by differentiating through the optimization problem itself. This approach has demonstrated it can sometimes reach more accurate solutions than first learning the unknown parameters and then solving the optimization problem. However, current end-to-end learning approach hardly scales up due to the complexity of solving and back-propagating through the optimization problem. The proposed approach is to also learn a (linear) reparameterization of the solutions, in low dimension, that leads to a much simpler surrogate optimization problem. Experimental results on 3 different domains (an advanced attacker/defender game on a network, a recommendation problem, and a portfolio optimization problem) validate the approach compared to both the 2-phase approach and the raw end-to-end approach.

Strengths: A good paper, a sensible approach, well described. And the experiments clearly and nicely make the point, in terms of performance and scalability. I had more trouble with Figure 7, though (especially 7b) ...

Weaknesses: My main comment here will be about linearity. The chosen reparameterization is linear, which I find limiting. I understand from the short discussion in the Appendix (though not even mentioned in the paper itself) that this is in order to preserve the good properties of the problem, but I would really be interested in the behavior on non-linear reparameterization in this context. Also, and this is another type of linear assumption you have to make, in section 4.3, you write "For simplicity, we assume our objective function f to be a linear function, and the feasible region S is compact, convex, and polyhedron." However, without these hypotheses, you couldn't apply Balghiti et al's results, could you? Do you think the result might still hold? In any case, I would remove the "For simplicity" from this sentence. Finally, the link between linearity (or 'quadraticity' of the optimization problem) and the behavior of the linear surrogate reparameterization also appear in the experimental results - as you point out - and would deserve deeper insights.

Correctness: The theoretical results seems OK: the first ones are straightforward, NS i had no time to dive deeper into Theorem 2, but it apparently heavily relies on similar result by Balghiti et al. [11], so it is mainly a matter of checking that the hypotheses are fulfilled. The empirical approach is described in detail. I didn't have the opportunity to run the provided code - hopefully some other reviewer will have done so - and anyway, providing the code is already a good sign :-)

Clarity: OK, and the clear figures did help a lot. Even though sometimes the conciseness imposed by the 8-pages limit results in needs to read some sentences or paragraphs several times.

Relation to Prior Work: Seems OK to me - but I might have myself missed some recent works in the area.

Reproducibility: Yes

Additional Feedback: I ticked "yes" for reproducibility because the code is provided. But you could not reproduce the experiments just from the text, as several important choices remain untold (e.g., which black-box optimizer - and parameters - is chosen from the scipy library). I am satisfied with the answers to my comments in the rebuttal, and that reinforces my "accept" recommendation.


Review 3

Summary and Contributions: This paper concerns the study of joint optimization of objective function and decision variable under a predict-then-optimize framework, assuming access to sampled data. The main proposal is to perform a reparametrization as a dimensionality reduction step so as to (1) speed up optimization; (2) improve generalization. The linear reparametrization perserves convexity for the optimization step and the subspace matrix can be learned along with the predictive model parameters by backpropagating through the KKT condition.

Strengths: The problem under study is very relevant to a lot of practical applications. The experiment section covers diverse use-cases and are very well-documented.

Weaknesses: Setup of the problem is relatively simplistic and extension from existing framework is more or less straightforward.

Correctness: The method and claims all look correct.

Clarity: Paper is well-motivated, clearly written and easy to follow. The schematics drawn are particularly helpful for understanding the problem.

Relation to Prior Work: The related work section surveys the prior works pretty thoroughly.

Reproducibility: Yes

Additional Feedback: \item Minor typo: Figure 3 label $y = Py \rightarrow x = Py$? \item Theorem 1 - is $P \geq 0$ constraint kind of arbitrary and potentially restrictive? \item Theorem 2 - would be easier to understand if the definition of $C$ can be written in math in the statement \item I couldn't quite follow the claim in line 157-159. ``For any hypothesis class, ..., this also preserves the convergence of learning the predictive model". It wasn't entirely clear to me if it converges/converges to what? \item The authors mentioned several times about smoothness property resulting from the parametrization, would be nice to see slightly more rigorous justification/elaboration. [Discussion Phase Addition: I would like to thank the authors for addressing my comments. Overall I think it's a relatively easy-to-follow framework and would like to leave my view unchanged.]


Review 4

Summary and Contributions: This paper proposed a linear reparameterization scheme as surrogate models for general optimization layers in the context of predict-then-optimize framework. Theoretical analysis of this framework was provided along several dimensions. - desirable properties of the optimization problem (convexity, sub-modularity) were retained under reparameterization. - Coordinate-wise quasiconvexity of the reparameterization matrix P was satisfied. - Sample complexity bounds of learning predictive model in surrogate problem were provided. Empirically on a set of three diverse domains (convex, non-convex, sub-modular) the proposed approach demonstrated significant advantages in both time complexity and decision quality.

Strengths: This paper deals with an predict-then-optimize problem in the framework of gradient-based end-to-end learning approaches in domains with optimization layers involved. Though only a simple linear reparameterization scheme is incorporated as surrogate model, the proposed approach is able to achieve better solution quality compared to two-stage learning, while avoiding the issues of non-smoothness of objective value and the problem of scalability. Theoretical analysis of linear reparameterization on complexity of solving the surrogate problem, learning the reparameterization and learning the predictive model are provided, which are sound and informative. Three different experiments cover three different types of objective functions. Evaluation on regret shows the better performance of the proposed approach, the demonstration of time consumption shows its efficiency, while visualization of the reparameterization provides a deeper understanding of how the surrogate captures the underlying problem structure.

Weaknesses: - In Theorem 2, both function f and function \phi are limited to linear functions, which is quite a special case and different from those used in experiments (convex, non-convex, sub modularity). An open question is whether Theorem 2 could be extended to these functions rather than only the linear one to some extent. - By using the linear reparameterization scheme, one needs to compute the derivatives with respect to the weights w and reparameterization matrix P, where P is a matrix and contains no less dimensions than the original variable x. In this view, it seems the time complexity is increased. - It would be better to explain how the parameters (learning rate, batch size, sample size, reparameterization size, etc.) in the experiments are chosen. In addition, adding experiment varying reparameterization size (e.g. from 10% to 100% of the problem size) would be more Informative for understanding the tradeoff between representational power and generalizability.

Correctness: The core idea of this paper is using linear transformation to optimize a surrogate model. Based on the idea, theorems and algorithms are derived clearly, which seem to be sound. Empirical part covers three different types of objective functions. Experimental settings are clearly explained.

Clarity: This paper is overall well written, except for some points that need to be further clarified. - In Theorem 1, P_i might be defined at first and the meaning of symbol \geq between a matrix P and a scaler 0 might be explained. - Explanation of the missing part of method DF when the number of nodes is beyond 60 in Figure 4. - Explanation of the method block in Figure 4. This method might also be introduced in line 177. Some typos: - In Figure 3, “ y=Py ”. - Line 196, “nodes en route”. - Additional symbol ‘]’ in equation 10 of supplementary materials.

Relation to Prior Work: Relation to prior work has been clearly discussed in related work section.

Reproducibility: Yes

Additional Feedback:

[Author Response · NeurIPS 2020]

**Reviewer #1**

We thank the reviewer for the constructive feedback. We will make the suggested clarifications and fix the typos.

The framework of the paper uses the model to improve the reparameterization directly. When the model is not specified or perhaps does not exist, using covariates as an alternative objective to optimize could be an extension of the current framework. Reparameterizing in such an extension is an interesting future direction to explore.

**Reviewer #2**

We thank the reviewer for the constructive feedback. We will clarify Figure 7. For 7(b), the intuition is that the learned reparameterizations put more weight on movies with higher average ratings (x-axis) and higher variation (y-axis) — the reparameterization is focused on distinguishing between different top-rated movies with some variance in opinions to specialize the recommendation set to the particular users.

**Linearity of reparameterization.** Whereas this paper shows that linear reparameterization provides a significant benefit, we agree that this opens the door to further research in reparameterization. In particular, exploring non-linear reparameterizations is an interesting future direction. It will require substantial additional development due to the resulting non-convex constraints, and the theoretical guarantees may not hold.

**Theorem 2.** We thank the reviewer for pointing out the confusion. We will remove the term "for simplicity" as recommended. Extending to convex objectives is an interesting topic for future work. It would allow us to adopt more flexible (e.g., convex) reparameterization while maintaining the theoretical guarantees.

**Reviewer #3**

We thank the reviewer for the constructive comments. We want to highlight that our approach should be understood in terms of how we reframe the predict-then-optimize problem in a conceptually different way. By doing optimization in a learned representation space instead of the problem's original space, we enable substantial benefits against the state-of-the-art approaches.

**Theorem 1.** Yes, Theorem 1 still holds without the condition $P \geq 0$. Throughout the paper, we assume the feasible region (if bounded) to be in the first quadrant, so that a non-negative reparameterization suffices. The reviewer is correct that the theorem and the proof still holds without this condition. We thank the reviewer for pointing this out and will clarify this in the write-up.

**Theorem 2.** The constant $C$ is defined as $C := sup_\theta(max_x f(x, \theta) - min_{x'} f(x', \theta))$. We will clarify this by adding a formal definition.

**Convergence of the predictive model.** When the objective function is linear and the hypothesis class has a finite Natarajan dimension, the generalization error will converge to 0 as the number of training examples approaches infinity. This indicates that the performance of the predictive model will converge to its expected performance too. (Technically, the parameters of the predictive model may not converge as they could alternate between two optimal solutions, but the performance converges asymptotically.) We will clarify this in the write-up.

**Reviewer #4**

We thank the reviewer for the constructive suggestions. We will make the suggested clarifications and fix the typos.

**Theorem 2.** We agree that extending to non-linear objective functions is an open and interesting question. In particular, our empirical results have shown that our reparameterization approach also works for non-linear objective functions. We think the theoretical result for a linear objective serves as an important step toward the convex case. The sample complexity of the linear case depends on the slope of the linear objective function (constant $C$ in Equation 4). An analogous term (e.g., Lipchitz constant) will likely appear in the convex case, so the result would likely be in terms of convex functions that are Lipschitz over the feasible region.

**Time complexity.** Using a smaller dimensional reparameterization reduces both the theoretical and empirical time complexity, despite having to learn and back-propagate through an additional parameter $P$. The reduced computational cost includes 1) inverting a smaller dimensional KKT matrix which takes roughly cubic less time 2) solving a lower dimensional optimization problem. The increased computational cost includes 1) matrix-vector multiplication $x = Py$ and 2) additional back-propagation to the parameter $P$, which is also matrix-vector multiplication and thus takes square time. Thus, overall the time complexity is reduced.

**Hyperparameters.** We hand-tuned the learning rate and reparameterization size for all competing methods. We will add more details about how the parameters are chosen to the appendix. We agree that adding an additional experiment varying reparameterization size would be informative for hyperparameter selection. We will also add this to the appendix.

[Meta-Review · NeurIPS 2020]

This was a unanimous accept.